# Long Term Survival of Pathological Thoracolumbar Fractures Treated with Vertebroplasty: Analysis Using a Nationwide Insurance Claim Database

**DOI:** 10.3390/jcm9010078

**Published:** 2019-12-27

**Authors:** Kuo-Yuan Huang, Shang-Chi Lee, Wen-Lung Liu, Jung-Der Wang

**Affiliations:** 1Department of Orthopedics, National Cheng Kung University Hospital, College of Medicine, National Cheng Kung University, Tainan 701, Taiwan; hkyuan@mail.ncku.edu.tw (K.-Y.H.); manga0522@gmail.com (W.-L.L.); 2Biostatistics Consulting Centre, National Cheng Kung University Hospital, College of Medicine, National Cheng Kung University, Tainan 701, Taiwan; chiling178@gmail.com; 3Department of Public Health, College of Medicine, National Cheng Kung University, Tainan 701, Taiwan

**Keywords:** thoracolumbar fracture, PMMA cement, pathological fracture, vertebroplasty, National Health Insurance Research Database

## Abstract

Background: There are still debates on the long-term outcome of treating pathological thoracolumbar fractures, including osteoporosis and oncologic problems, using vertebroplasty. Methods: We collected 8625 patients with pathological thoracolumbar fractures (ICD-9-CM codes 733.13 combined with 805.2 or 805.4) between the years of 2003 to 2013, from the two million random samples from the National Health Insurance Research Database in Taiwan. Survival analysis was conducted to estimate the mortality risks of different treatments, including vertebroplasty (*n* = 1389), conventional open surgery (*n* = 1219), or conservative treatment (*n* = 6017). A multivariable Cox proportional hazard model was constructed for adjustment of age, gender, comorbidities and complications. Results: Crude incidence rate of patients with pathological thoracolumbar fractures in Taiwan gradually increased year by year. Compared with conservative treatment, conventional open surgery and vertebroplasty seemed to improve long-term survival with adjusted hazard ratios (aHR) of 0.80 (95% confidence interval (CI) 0.70–0.93), and 0.87 (95% CI 0.77–0.99), respectively. The survival advantage of vertebroplasty appeared more evident for those aged over 75. However, we were unable to rule out confounding by indication. Conclusion: Although conventional open surgery would usually be the best choice for the treatment of patients with pathological thoracolumbar fractures, database information from current real-world practice appears to support vertebroplasty as a viable choice for elderly people over 75 years of age.

## 1. Introduction

Patients with pathological thoracolumbar fractures, including osteoporosis and oncologic problems, usually suffer from mechanical back pain, neurological deficits, spinal deformity and poor quality of life [1,2,3]. For quick pain relief, vertebroplasty has been applied in these conditions, including osteoporosis, metastasis, and multiple myeloma [1,2,4,5]. However, conventional open surgery such as anterior or posterior decompression or stabilization, or fusion surgery, has been indicated for patients with pathological thoracolumbar fractures who have progressive motor weakness, cauda equina syndrome, or spinal instability [6,7,8,9]. Polymethylmethacrylate (PMMA) cement is the most commonly used material for augmentation of fractured vertebral bodies to relieve debilitating back pain [10]. The majority of patients with osteoporotic compression fractures receiving PMMA vertebroplasty seemed to show excellent results on back pain relief rapidly in short-term follow-up [2,11,12]. This procedure can help elderly patients quickly return to daily activity and possibly decrease the social cost of family care and the morbidities of being bedridden in the long term [13]. On the other hand, the non-absorbent and non-biodegradable material properties of PMMA cement, as well as the release of high temperatures during polymerization, might result in necrosis, fibrosis, lack of revascularization, and inhibition of the natural repair process around the PMMA cement within the fractured vertebral body in patients with failed PMMA vertebroplasty [14]. In addition, there is still controversy over the beneficial effect of the treatment of osteoporotic thoracolumbar fractures [15,16,17,18]. 

Kyphoplasty, another cement augmentation technique, which uses a balloon or other instrument to reduce the collapsed vertebrae and then injects the cement into the created cavity, was developed to minimize the complication of cement leakage [19]. Some researchers have explored the long-term mortality and survival for the osteoporotic vertebral compression fractures treated with vertebroplasty or kyphoplasty compared with conservative treatment, and most found kyphoplasty had better survival than vertebroplasty, and both kyphoplasty and vertebroplasty seemed to have better survival than conservative treatment [20,21,22,23,24,25]. However, there is still limited research [9] comparing the long-term survival of pathological thoracolumbar fractures treated with vertebroplasty, conventional open surgery, and conservative treatment, as well as their related complications including: pulmonary embolism [26]; leakage of cement causing neurological deficits [27], shock [28] or death [29]; vertebral infection or osteomyelitis [30,31]; adjacent vertebral fractures [32]; and loosening of PMMA cement [33]. 

Since kyphoplasty was introduced much later than vertebroplasty and its reimbursement by the National Health Insurance (NHI) of Taiwan has not yet been formally approved, the sample size and follow-up period would be too small for research. Thus, we designed a retrospective cohort study to compare the long-term survival and complications of the patients with pathological thoracolumbar fractures who received vertebroplasty, conventional open spinal surgery, or conservative treatments.

## 2. Material and Methods

### 2.1. Patient Subjects

We retrieved and analyzed the data file of 2 million random samples from the National Health Insurance, of which the sampled cohorts were followed for 11 years (2003–2013). We enrolled patients with pathological thoracolumbar fractures, including osteoporosis and oncologic problems, from hospitalized data files according to the ICD-9-CM codes 733.13 and 805.2, or 733.13 and 805.4. These diagnostic codes, which were used in our reimbursement of the NHI, were relatively broad and we were unable to classify them into detailed morphological or pathological types. In contrast to those receiving open surgery or conservative treatment, we found most patients receiving vertebroplasty were above 60 years old. Therefore, we stratified the age group of the patients above 60 into 5 year intervals. The data files were further stratified into 3 categories, as summarized in Figure 1: those receiving vertebroplasty (*n* = 1389), conventional open surgery (*n* = 1219) or conservative treatment (*n* = 6017) according to the presence of operation codes. All patients were followed up until death or censored by the end of 2013. According to the guideline for the prescription of anti-osteoporotic drugs in the NHI of Taiwan, since 2011 they were only reimbursed after patients met the criteria of osteoporosis in a bone mineral density examination. Thus, most patients with the diagnosis of osteoporosis before 2011 usually did not receive a bone mineral density test and we were unable to validate them. Alternatively, we chose to study the incidence rate, comorbidities, and mortality rate of patients with a diagnosis of pathological thoracolumbar fractures, and their related complications, such as pulmonary embolism, vertebral osteomyelitis or infections, adjacent fracture, refracture or other vertebral fracture, leakage of cement causing neurological deficits, shock or death, and loosening of PMMA cement, were analyzed.

The short-term related medical expenses and overall days of hospitalization among the three groups were also compared.

### 2.2. Statistics

Statistical analysis was performed at the Health and Welfare Data Science Collaboration Centers of the Ministry of Health and Welfare in Taiwan. We conducted a Kaplan–Meier survival analysis and multivariable Cox proportional hazards model to estimate the mortality risks of different treatments after adjustment for age, gender, comorbidities and complications. 

## 3. Results

The annual crude incidence rates of pathological thoracolumbar fractures among females are generally higher than those of males in Taiwan (Appendix A). From 2003 to 2013, the group receiving vertebroplasty grew while those receiving conservative treatment declined (Appendix A). We also found that the older the age, the higher the proportions were for choosing vertebroplasty or conservative treatment, but there was no such trend in people receiving conventional open surgery. Namely, there were a higher proportion of patients receiving vertebroplasty recently, and their average follow-up time was shorter compared with those receiving conservative treatment.

Comorbidities and complications were also analyzed, which are summarized in Table 1. We found the incidence of related complications, including pulmonary embolism, vertebral osteomyelitis or infections, adjacent fracture, refracture, or other vertebral fracture, were not increased in the groups of vertebroplasty and conventional open surgery. Furthermore, the numbers of other complications such as shock, death, or loosening of PMMA cement were too small to be further analyzed (*n* ≤ 4). The leakage of cement causing neurological deficits were not included in our study due to the lack of a corresponding diagnostic code specific to cement leakage.

The survival analysis by Kaplan–Meier method is summarized in Appendix A. Because both the conservative treatment and conventional open surgery group contained higher proportions of patients who were under 59 years old at the time of fracture, we reanalyzed the survival function limited to patients older than 75, and found that the survival of vertebroplasty was significantly higher than those receiving conservative treatment (log rank test, *p* value = 0.0002, in Figure 2). Table 2 summarizes the results of the Cox proportional hazard model after adjustments for age, gender, comorbidities and complications. We found that patients having pathological thoracolumbar fractures with comorbidities of certain cancer types, kidney failure, stroke, diabetes mellitus, chronic obstructive pulmonary disease (COPD) and hip fractures had higher risk of mortality. Survival improved in both conventional open surgery and vertebroplasty compared to conservative treatment with adjusted hazard ratios (aHR) of 0.80 [95% confidence interval (CI) 0.70–0.93], and 0.87 (95% CI 0.77–0.99), respectively.

We also compared hospitalization length and medical expenses. In the conservative treatment group, there were 3885 patients (64.6%) who were hospitalized for less than 7 days; however, in the vertebroplasty group, 933 patients (67.2%) were hospitalized for less than 7 days; in the conventional open surgery group, 546 patients (44.8%) were hospitalized for less than 7 days, while 436 patients (35.8%) were hospitalized for 8–14 days. The median medical expenses were 670, 1237, 2985 US dollars in the conservative treatment, vertebroplasty, and conventional open surgery groups, respectively. Thus, patients with vertebroplasty generally showed a shorter stay of hospitalization and lower medical expense compared with those receiving conventional open surgery (Appendix A).

## 4. Discussion

From 2003 to 2013, people with pathological thoracolumbar fractures receiving vertebroplasty grew while those receiving conservative treatment declined (Appendix A). This may be related to the progress in the technique of the vertebroplasty operation and the reimbursement policy. Given an alternative choice for pain relief [2,12], one would be concerned with whether it improves long-term survival and reduces complications compared with conventional open surgery and conservative treatment. The initial survival analysis of all patients using the Kaplan–Meier method included all patients and did not detect any significant difference between vertebroplasty and conservative treatment (Appendix A). As there was a higher proportion of younger patients in both the conservative treatment and conventional open surgery group in comparison with vertebroplasty, we limited the survival analysis to patients older than 75. While there is no difference between vertebroplasty and conventional open surgery, the 11-year survival of patients who received vertebroplasty was significantly higher than those who received conservative treatment (log rank test, *p* value = 0.0002), as summarized in Figure 2. After controlling for potential confounders of age, sex, and major comorbidities that might lead to premature mortality, such as cancers, renal failure, stroke, diabetes, COPD, hip fracture, pulmonary embolism [34], and osteomyelitis, the Cox models showed that conventional open surgery significantly improved survival compared with conservative treatment (adjusted hazard ratio (aHR) = 0.80 with 95% confidence interval (CI) 0.70–0.93). Patients with pathological thoracolumbar fractures who received vertebroplasty seemed to survive longer than those receiving conservative treatment, and the aHR of vertebroplasty was 0.87 (95% CI 0.77–0.99), which appeared more evident for patients aged over 75 years old. However, since the choice of treatment was usually based on the fracture type, anatomical factors, health status etc., and this study was not a prospective randomized controlled trial, we are unable to completely rule out confounding by indication. Namely, patients who received conservative treatment could differ in nature from those undergoing conventional open surgery or vertebroplasty procedure, which may not be completely controlled by adjustments in statistical modelling. Therefore, we tentatively conclude that current real world practice of vertebroplasty in Taiwan seems to show a better long-term survival for those receiving vertebroplasty than conservative treatment, especially among those older than 75 years of age.

In the comorbidity analysis, we found all cancer disclosed an aHR nearly 2–3 times greater than no cancer, except for colon cancer. This may be partially explained by the fact that colon cancer that is treated early (especially at Stage I) shows little or no loss of life expectancy [35,36] and Taiwan has had a national screening campaign for more than a decade. Patients with kidney failure who receive regular hemodialysis are prone to osteoporotic fractures accompanied by increased morbidity and mortality [37], and this could explain why the aHR increased up to 1.96. COPD is associated with osteoporosis, vertebral compression fracture, and increased mortality [38], which could account for an aHR of 1.61. In fact, the list of statistically significant comorbidities in Table 2 corroborates with current existing knowledge, which in turn partially validates our study.

There were the following limitations in this study. First, the ICD codes we enrolled as “pathological” fractures were relatively broad, as these data were abstracted from the claim database. They usually contain various types of fractures resulted from osteoporosis and oncologic problems. Table 1 showed that only less than 5% were cancer-related pathological fractures. Since patients with cancer can be waived from copayment in the Taiwan NHI, such a diagnosis is generally preserved in the five major codes in the claim database. In contrast, osteoporosis tends to be omitted and underestimated if the patient has more than five chronic diseases. Second, as our current reimbursement scheme in the National Health Insurance does not include kyphoplasty, we were unable to assess the long-term outcome for this treatment option for pathological thoracolumbar fractures. Although some research showed both vertebroplasty and kyphoplasty were better than conservative treatment or sham procedures without cement, and kyphoplasty may be better than vertebroplasty in restoring the vertebral height [39] and possibly giving better survival [3], other studies [40,41] did not show consistent results, especially on patients with compression fractures. Future studies are warranted to explore this unresolved issue. Third, because only up to five diagnoses were recorded in the coding of our national datasets of reimbursement, some minor complications related to vertebroplasty might not have been included, such as small leakages of injected cement, or adjacent vertebral fracture. However, major complications related to vertebroplasty which would usually affect long-term survival, such as pulmonary embolism and/or major comorbidities, were already included in the multivariable Cox model (Table 2). Thus, the lack of information on detailed minor complications in the database does not seem to bias our final outcome related to long-term survival. Fourth, the National Health Insurance Research Database does not contain details on the cement material used in vertebroplasty. Because PMMA is regularly reimbursed by the Taiwan National Health Insurance without copayment, we believe that most vertebroplasty conducted in Taiwan would use PMMA. However, we were unable to make more inference about detailed techniques, including content [11], viscosity of cement [42] and/or infusion rate [11]. Future studies are warranted to explore these detailed contents and procedures, including the number of levels performed, etc. Fifth, although vertebroplasty is less invasive, and our clinical observation found that elderly patients with thoracolumbar fracture recovered faster after vertebroplasty with a shorter stay of hospitalization and lower medical expense compared with conventional anterior and posterior spinal decompression or fusion surgeries (Appendix A), we did not collect the data on patients’ quality of life and are unable to make any inference on how it improves patient’s function and quality of life. Future studies are warranted to clarify whether vertebroplasty would also be beneficial to quality of life.

## 5. Conclusions

In Taiwan, patients aged over 75 with pathological thoracolumbar fractures receiving vertebroplasty had a better survival compared with those receiving conservative treatment. Moreover, vertebroplasty still showed an improved survival rate in comparison with conservative treatment with an aHR of 0.87 after adjustment of age, sex, demographic and major clinical risk factors. Since we were unable to rule out confounding by indication, we conclude that the overall current practice of vertebroplasty in Taiwan seems acceptable or beneficial for long-term survival. More studies are needed to determine if vertebroplasty would improve quality of life.

## Figures and Tables

**Figure 1 jcm-09-00078-f001:**
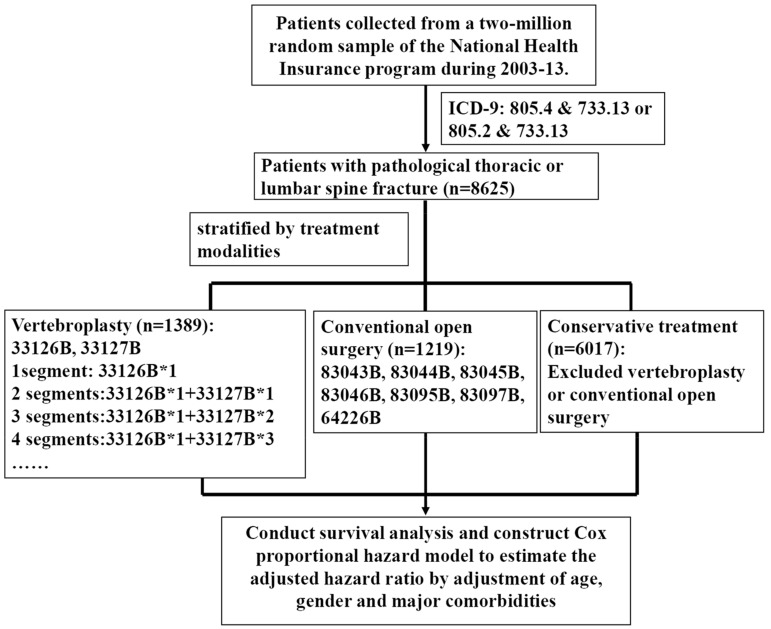
Schematic diagram of the selection of study subjects.

**Figure 2 jcm-09-00078-f002:**
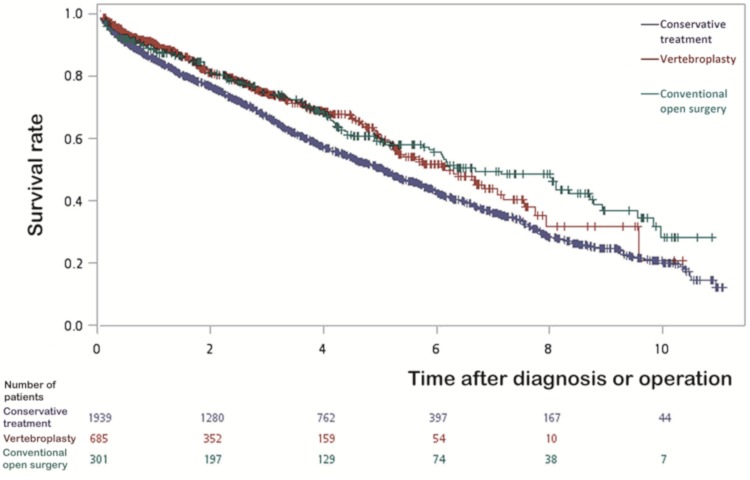
Kaplan–Meier survival curves of different treatments for patients whose ages were more than 75 years old. The y-axis indicates the survival rate, and the x-axis represents the time after diagnosis (or operation), while the frequencies on the bottom show the number of patients in each group that still survived up to that time point. Log rank test analysis showed a significant difference (*p* = 0.0002) between conservative treatment and vertebroplasty, while there was no significant difference (*p* = 0.64) between vertebroplasty and conventional open surgery.

**Table 1 jcm-09-00078-t001:** Demographic and clinical characteristics of included patients stratified by conservative treatment, vertebroplasty and conventional open surgery.

Treatment	Conservative Treatment (*n* = 6017)	Vertebroplasty (*n* = 1389)	Conventional Open Surgery (*n* = 1219)
Gender ** (%Males)	44.8	25.1	43.4
Age ** < 59 (%)	40.1	16.8	45.1
60–64 (%)	7.0	7.5	7.5
65–69 (%)	8.6	10.4	10.2
70–74 (%)	12.1	15.9	12.6
75–79 (%)	12.8	20.9	12.4
>79 (%)	19.4	28.4	12.3
No. of deceased ** (%)	26.2	19.2	17.4
Comorbidity:			
All cancer (%)	2.6	2.5	1.6
Lung cancer (%)	0.5	0.5	0.5
Liver cancer (%)	0.4	0.5	0.5
Colon cancer (%)	0.3	0.4	0.4
Kidney failure (%)	1.7	1.7	1.2
Stroke * (%)	4.6	3.5	2.9
Diabetes mellitus (%)	4.0	3.3	5.0
Autoimmunity disease (%)	0.5	0.8	0.5
COPD ** (%)	2.8	0.9	1.8
Hip fracture (%)	2.5	2.0	1.6
Complication			
Pulmonary embolism (%)	0.7	0.4	≦0.3
Vertebral osteomyelitis or infections (%)	0.9	1.0	1.0
Adjacent fracture, refracture or other vertebral fracture (%)	<0.05	<0.25	<0.25
Follow-up time ** (years)	4.8 ± 3.2	3.2 ± 2.5	4.7 ± 3.1

*: *p* < 0.05; **: *p* < 0.0001; COPD: chronic obstructive pulmonary disease.

**Table 2 jcm-09-00078-t002:** Hazard ratio (HR), adjusted HR (aHR), and 95% confidence interval (CI) of prognostic factors estimated from Cox proportional hazard model.

Variables	Simple Cox Model		Multiple Cox Model	
	HR	95% CI	aHR	95% CI
Gender (M/F)	1.02	(0.93–1.11)	1.45	(1.32–1.58)
Age < 59	1		1	
60–64	2.11	(1.63–2.74)	2.23	(1.71–2.89)
65–69	3.02	(2.44–3.73)	3.24	(2.61–4.02)
70–74	4.81	(4.04–5.73)	5.06	(4.23–6.06)
75–79	7.57	(6.44–8.90)	8.10	(6.85–9.58)
>79	13.71	(11.80–15.94)	13.87	(11.86–16.21)
Conservative treatment	1		1	
Vertebroplasty	1.07	(0.94–1.22)	0.87	(0.77–0.99)
Conventional open surgery	0.68	(0.59–0.79)	0.80	(0.70–0.93)
Comorbidity				
All cancer	4.43	(3.73–5.27)	2.20	(1.72–2.82)
Lung cancer	8.40	(6.09–11.59)	2.80	(1.88–4.16)
Liver cancer	7.33	(5.11–10.52)	2.70	(1.77–4.11)
Colon cancer	2.63	(1.58–4.37)	0.55	(0.31–0.95)
Kidney failure	3.50	(2.79–4.40)	1.96	(1.55–2.46)
Stroke	2.16	(1.84–2.53)	1.17	(1.00–1.38)
Diabetes mellitus	1.55	(1.30–1.85)	1.11	(0.93–1.33)
Autoimmunity disease	0.91	(0.52–1.61)	0.73	(0.41–1.29)
COPD	3.85	(3.25–4.57)	1.61	(1.35–1.92)
Hip fracture	2.59	(2.13–3.15)	1.43	(1.17–1.75)
Complication				
Pulmonary embolism	3.44	(2.37–4.99)	1.81	(1.24–2.63)
Vertebral osteomyelitis or infections	1.98	(1.41–2.78)	1.73	(1.23–2.44)

COPD: chronic obstructive pulmonary disease.

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
