# Peer review of "Long Term Survival of Pathological Thoracolumbar Fractures Treated with Vertebroplasty: Analysis Using a Nationwide Insurance Claim Database"

_jcm, 2019, doi:10.3390/jcm9010078_

Round 1

Reviewer 1 Report

In this manuscript, the authors investigated the long-term survival and complications of patients with “pathological thoracolumbar fracture” and compared three treatment options: conservative treatment, vertebroplasty, and open surgery. They concluded that vertebroplasty would be a viable option for elderly people with this kind of fracture. I agree with the authors’ opinion shown in the conclusion, however, this study seems contain serious flaws and, for me, the results do not support their conclusion. I have following concerns:

1. The definition of “pathological” appears unclear. It might include osteoporotic fracture or can be solely fractures associated with oncologic problem. According to the references, they mentioned it seems include osteoporosis. It should be stated clearly. The ICD codes they referred were very broad meaning and may contain various types of fractures. It seems difficult to define “pathological” only with the definitions of these codes.
2. As the authors mentioned, the proportion of patients age group <59 were proportionally high in conservative and open surgery groups. I am wondering if patients under 59 really had pathological fractures in common meaning. Also, this age group should be divided into multiple categories.
3. As the authors mentioned, the rationales for treatment choice were unclear. Even with the adjustment of some cofounders, the best treatment varies by injury type. For instance, open surgery should be indicated severe burst fractures with instability even without neurological deficit and likely have worse prognosis compared with simple vertebral fracture treated with vertebroplasty. Analysis without this information means nothing.

Author Response

Reviewer #1

The definition of “pathological” appears unclear. It might include osteoporotic fracture or can be solely fractures associated with oncologic problem. According to the references, they mentioned it seems include osteoporosis. It should be stated clearly. The ICD codes they referred were very broad meaning and may contain various types of fractures. It seems difficult to define “pathological” only with the definitions of these codes.

REPLY: As suggested, we have tried to clarify the definition of pathological fracture in the revised manuscript, as follows:  

Section of Material and Methods (Lines 75-79):

We enrolled patients with pathological thoracolumbar fractures, including osteoporosis and oncologic problem, from hospitalized data files according to the ICD-9-CM code: 733.13 and 805.2, or, 733.13 and 805.4. These diagnostic codes, which were used in our reimbursement of NHI, were relatively broad and we were unable to classify into detailed morphological or pathological types.

Discussion section at the paragraph of limitations (Lines 195-201):

There were the following limitations in this study: First, The ICD codes we enrolled as “pathological” fracture were relatively broad, as these data were abstracted from claim database. They usually contain various types of fractures resulted from osteoporosis and oncologic problems. Table 1 showed that only less than 5% were cancer related pathological fracture. Since patients with cancer can be waived from co-payment in the Taiwan NHI, such a diagnosis is generally preserved in the 5 major codes in the claim database. In contrast, osteoporosis tends to be omitted and underestimated if the patient has more than 5 chronic diseases.

As the authors mentioned, the proportion of patients age group <59 were proportionally high in conservative and open surgery groups. I am wondering if patients under 59 really had pathological fractures in common meaning. Also, this age group should be divided into multiple categories.

REPLY: Thanks for your suggestion. As we have responded to your previous question and added the limitation paragraph, our claim data did not contain detailed information of morphological and/or pathological fractures, we are unable to further classify or divided into subgroups. And we have revised some sentences, as follows:

Section of Material and Methods (Lines 79-81):

In contrast to those receiving open surgery or conservative treatment, we found most patients receiving vertebroplasty were above 60 years old. Therefore, we stratified patients’ age group above 60 in 5-year interval.

As the authors mentioned, the rationales for treatment choice were unclear. Even with the adjustment of some cofounders, the best treatment varies by injury type. For instance, open surgery should be indicated severe burst fractures with instability even without neurological deficit and likely have worse prognosis compared with simple vertebral fracture treated with vertebroplasty. Analysis without this information means nothing.

REPLY: Thanks for your constructive comments. Unfortunately, our claim database does not contain such detailed information about fracture types, anatomical factors, and health status, etc., and we are unable to make further stratification. Thus, we made a conservative conclusion and related limitations were discussed in our revised manuscript, as follows:

Discussion section, near the end of the 1st paragraph (Lines 170-185):

After controlling for potential confounders of age, sex, and major co-morbidities that might lead to premature mortality, such as cancers, renal failure, stroke, diabetes, COPD, hip fracture, pulmonary embolism [34], and osteomyelitis, etc., the Cox models showed that conventional open surgery significantly improved survival compared to conservative treatment (adjusted Hazard Ratio (aHR)=0.80 with 95% confidence interval (CI) 0.70-0.93. Patients with pathological thoracolumbar fractures who received vertebroplasty seemed to survive longer than those receiving conservative treatment, and the aHR of vertebroplasty was 0.87 (95% CI 0.77-0.99), which appeared more evident for patients aged over 75 years old. However, since the choice of treatment was usually based on the fracture type, anatomical factors, health status etc. and this study was not a prospective randomized controlled trial, we are unable to completely rule out confounding by indication. Namely, patients who received conservative treatment could differ in nature from those undergoing conventional open surgery or vertebroplasty procedure, which may not be completely controlled by adjustments in statistical modelling. Therefore, we tentatively concluded that current real world practice of vertebroplasty in Taiwan seems to show a better long-term survival for those receiving vertebroplasty than conservative treatment, especially among those older than 75 years of age.

Discussion section at the paragraph of limitations (Lines 195-201):

There were the following limitations in this study: First, The ICD codes we enrolled as “pathological” fracture were relatively broad, as these data were abstracted from claim database. They usually contain various types of fractures resulted from osteoporosis and oncologic problems. Table 1 showed that only less than 5% were cancer related pathological fracture. Since patients with cancer can be waived from co-payment in the Taiwan NHI, such a diagnosis is generally preserved in the 5 major codes in the claim database. In contrast, osteoporosis tends to be omitted and underestimated if the patient has more than 5 chronic diseases.

Reviewer 2 Report

I applaud the authors for taking on this analysis, and the study takes on an interesting subject. However, there are some major concerns.

Major

While recognized and mentioned by the authors that they could not correct for indication, this remains a fundamental and important point missing from the analysis. The real question is for the same pathology and presentation is vertebroplaty, surgery or conservative treatment more optimal. We cannot address this question with the data. The type of treatment like surgery likely was based on the fracture type, anatomical factors, health status etc. While the authors have corrected for some major factors like cancer types, the lack also of whether osteoporosis was present is another important cofactor.   increased survival of the surgery group may also be the case simply because patients with who were chosen for an operation were healthier and with less comorbidities.

Minor

Table 1 - you indicate “low” for adjacent fracture what does this mean, these areas are supposed to present a %.

Author Response

Reviewer #2:

While recognized and mentioned by the authors that they could not correct for indication, this remains a fundamental and important point missing from the analysis. The real question is for the same pathology and presentation is vertebroplaty, surgery or conservative treatment more optimal. We cannot address this question with the data. The type of treatment like surgery likely was based on the fracture type, anatomical factors, health status etc. While the authors have corrected for some major factors like cancer types, the lack also of whether osteoporosis was present is another important cofactor. increased survival of the surgery group may also be the case simply because patients with who were chosen for an operation were healthier and with less comorbidities.

REPLY: Thanks for your constructive comments. Unfortunately, our claim database does not contain such detailed information about fracture types, anatomical factors, and health status, etc., and we are unable to make further stratification. Thus, we made a conservative conclusion and related limitations were discussed in our revised manuscript, as follows:

Discussion section, near the end of the 1st paragraph (Lines 170-185):

After controlling for potential confounders of age, sex, and major co-morbidities that might lead to premature mortality, such as cancers, renal failure, stroke, diabetes, COPD, hip fracture, pulmonary embolism [34], and osteomyelitis, etc., the Cox models showed that conventional open surgery significantly improved survival compared to conservative treatment (adjusted Hazard Ratio (aHR)=0.80 with 95% confidence interval (CI) 0.70-0.93. Patients with pathological thoracolumbar fractures who received vertebroplasty seemed to survive longer than those receiving conservative treatment, and the aHR of vertebroplasty was 0.87 (95% CI 0.77-0.99), which appeared more evident for patients aged over 75 years old. However, since the choice of treatment was usually based on the fracture type, anatomical factors, health status etc. and this study was not a prospective randomized controlled trial, we are unable to completely rule out confounding by indication. Namely, patients who received conservative treatment could differ in nature from those undergoing conventional open surgery or vertebroplasty procedure, which may not be completely controlled by adjustments in statistical modelling. Therefore, we tentatively concluded that current real world practice of vertebroplasty in Taiwan seems to show a better long-term survival for those receiving vertebroplasty than conservative treatment, especially among those older than 75 years of age.

Discussion section at the paragraph of limitations (Lines 195-201):

There were the following limitations in this study: First, The ICD codes we enrolled as “pathological” fracture were relatively broad, as these data were abstracted from claim database.  They usually contain various types of fractures resulted from osteoporosis and oncologic problems. Table 1 showed that only less than 5% were cancer related pathological fracture. Since patients with cancer can be waived from co-payment in the Taiwan NHI, such a diagnosis is generally preserved in the 5 major codes in the claim database. In contrast, osteoporosis tends to be omitted and underestimated if the patient has more than 5 chronic diseases.

Table 1 - you indicate “low” for adjacent fracture what does this mean, these areas are supposed to present a %.

REPLY: Thanks. The statistical center of the Ministry of Health and Welfare forbids any number of cases smaller than 3 to be brought out, we simply indicated as “low” in our previous presentation of Table 1. Following your constructive advice, we have calculated the lowest percentages for incidence rate of adjacent fracture, refracture and other vertebral fracture, which were <0.05%, <0.25%, and <0.25%, respectively, for conservative treatment, vertebroplasty, and open surgery, as modified in Table 1:

Table 1. Demographic and clinical characteristics of included patients stratified by conservative treatment, vertebroplasty and conventional open surgery.

Treatment

Conservative treatment (N=6017)

Vertebroplasty (N=1389)

Conventional open surgery (N=1219)

Gender** (%Males)

44.8

25.1

43.4

Age**<59 (%)

40.1

16.8

45.1

60-64 (%)

7.0

7.5

7.5

65-69 (%)

8.6

10.4

10.2

70-74 (%)

12.1

15.9

12.6

75-79 (%)

12.8

20.9

12.4

>79 (%)

19.4

28.4

12.3

No. of deceased** (%)

26.2

19.2

17.4

Co-morbidity:

All Cancer (%)

2.6

2.5

1.6

Lung Cancer (%)

0.5

0.5

0.5

Liver Cancer (%)

0.4

0.5

0.5

Colon Cancer (%)

0.3

0.4

0.4

Kidney failure (%)

1.7

1.7

1.2

Stroke* (%)

4.6

3.5

2.9

Diabetes Mellitus (%)

4.0

3.3

5.0

Autoimmunity disease (%)

0.5

0.8

0.5

COPD** (%)

2.8

0.9

1.8

Hip fracture (%)

2.5

2.0

1.6

Complication

Pulmonary Embolism (%)

0.7

0.4

≦0.3

Vertebral osteomyelitis or infections (%)

0.9

1.0

1.0

Adjacent fracture, refracture or other vertebral fracture (%)

<0.05

<0.25

<0.25

Follow-up time** (yrs)

4.8±3.2

3.2±2.5

4.7±3.1

*: p<0.05; **: p<0.0001; COPD: chronic obstructive pulmonary disease.

Reviewer 3 Report

Reasonable study of 1389 vertebroplasties, 1219 open surgeries, and 6017 conservative treatments using ICD-9 codes for pathologic thoracolumbar fracture from the National Health Insurance Research Database in Taiwan. The authors find that open surgery and vertebroplasty improve long term survival with aHR 0.80 and 0.87 respectively, and more so for those aged >75. 

Comments below:

1) The rates of malignancy and osteomyelitis are quite low in the dataset. Are the majority of pathologic fractures due to osteoporosis? This should be stated.

2) Did the authors stratify by # of vertebral levels? May be an important predictor of outcome.

3) Why was the follow-up time lower for those with vertebroplasties?

4) Due to their comparatively small sample sizes, may be of interest to do a subgroup analysis of those with malignancy and infection vs. osteoporosis and evaluate whether outcomes differ between the surgery, vertebroplasty and conservative management.

Author Response

Reviewer #3:

The rates of malignancy and osteomyelitis are quite low in the dataset. Are the majority of pathologic fractures due to osteoporosis? This should be stated.

REPLY: Thanks. The rates of malignancy would be what we have showed in Table 1, or, generally less than 5%. Since patients with cancer can be waived from co-payment in the Taiwan NHI, such a diagnosis is generally preserved in the 5 major codes in the claim database for hospitalized patients. However, the condition for osteoporosis is different or generally underestimated. According to the guideline for the prescription of anti-osteoporotic drug in NHI of Taiwan, it was reimbursed only after patients met the criteria of osteoporosis in the bone mineral density examination since 2011. Thus, most patients with the diagnosis of osteoporosis before 2011 usually did not receive bone mineral density test and we were unable to validate it, though probably most of these patients belonged to this category.
We have revised the manuscript, as follows:

Section of Material and Methods (Lines 75-79):

We enrolled patients with pathological thoracolumbar fractures, including osteoporosis and oncologic problem, from hospitalized data files according to the ICD-9-CM code: 733.13 and 805.2, or, 733.13 and 805.4. These diagnostic codes, which were used in our reimbursement of NHI, were relatively broad and we were unable to classify into detailed morphological or pathological types.

Discussion section at the paragraph of limitations (Lines 195-201):

There were the following limitations in this study: First, The ICD codes we enrolled as “pathological” fracture were relatively broad, as these data were abstracted from claim database. They usually contain various types of fractures resulted from osteoporosis and oncologic problems. Table 1 showed that only less than 5% were cancer related pathological fracture. Since patients with cancer can be waived from co-payment in the Taiwan NHI, such a diagnosis is generally preserved in the 5 major codes in the claim database. In contrast, osteoporosis tends to be omitted and underestimated if the patient has more than 5 chronic diseases.

Did the authors stratify by # of vertebral levels? May be an important predictor of outcome.

REPLY: Thanks for your kind suggestion. Unfortunately, this hypothesis was not included in our original list, as our main aim is to compare the long-term survival among three different types of management. The Figure 1 simply indicates that we have included all patients with vertebroplasty of one and multiple levels or segments. In general, it takes about 15-20 minutes to perform one level of vertebroplasty, while 2 levels would take about double operation duration, etc. Although such a short increase in duration and procedure usually would not affect too much about long-term survival, this predictor would be of interest for future exploration into details of operation procedures. Thus, we have added the above point in our Discussion section, as follows:

Discussion section (Lines 215-221):

Fourth, the National Health Insurance Research Database does not contain details on the cement material used in vertebroplasty. Because PMMA is regularly reimbursed by the Taiwan National health insurance without co-payment, we believe that most vertebroplasty conducted in Taiwan would use PMMA. However, we were unable to make more inference about detailed techniques, including content [11] and viscosity of cement [42] and/or infusion rate [11]. Future studies are warranted to explore into these detailed contents and procedures, including the number of levels performed, etc.

Why was the follow-up time lower for those with vertebroplasties?

REPLY: Thanks for your kind enquiry. From 2003 to 2013, the group receiving vertebroplasty grew while those receiving conservative treatment declined (Supplementary Fig. 2, as is also shown at the end of this response), and there was no such trend in people receiving conventional open surgery. Therefore, there were a higher proportion of patients receiving vertebroplasty recently, and their average follow-up time was shorter compared with those receiving conservative treatment.

We have revised the manuscript, as follows:

Results Section (Lines, 105-110):
From 2003 to 2013, the group receiving vertebroplasty grew while those receiving conservative treatment declined (Fig. S2). We also found that the older the age, the higher the proportions were for choosing vertebroplasty or conservative treatment. But there is no such trend in people receiving conventional open surgery. Namely, there were a higher proportion of patients receiving vertebroplasty recently, and their average follow-up time was shorter compared with those receiving conservative treatment.

Due to their comparatively small sample sizes, may be of interest to do a subgroup analysis of those with malignancy and infection vs. osteoporosis and evaluate whether outcomes differ between the surgery, vertebroplasty and conservative management.

REPLY: Thanks for your kind suggestion. It’s worthwhile and of interest to do an analysis for those with malignancy and infection vs. osteoporosis and to evaluate whether long-term outcomes differ among 3 types of management: open surgery, vertebroplasty, and conservative treatment. However, as our claim data did not contain detailed clinical information, we were unable to further classify or divided into subgroups. And we have revised our manuscript to acknowledge this limitation, as follows:

Section of Material and Methods (Lines 75-79):

We enrolled patients with pathological thoracolumbar fractures, including osteoporosis and oncologic problem, from hospitalized data files according to the ICD-9-CM code: 733.13 and 805.2, or, 733.13 and 805.4. These diagnostic codes, which were used in our reimbursement of NHI, were relatively broad and we were unable to classify into detailed morphological or pathological types.

Discussion section at the paragraph of limitations (Lines 195-201):

There were the following limitations in this study: First, The ICD codes we enrolled as “pathological” fracture were relatively broad, as these data were abstracted from claim database. They usually contain various types of fractures resulted from osteoporosis and oncologic problems. Table 1 showed that only less than 5% were cancer related pathological fracture. Since patients with cancer can be waived from co-payment in the Taiwan NHI, such a diagnosis is generally preserved in the 5 major codes in the claim database. In contrast, osteoporosis tends to be omitted and underestimated if the patient has more than 5 chronic diseases.

Round 2

Reviewer 1 Report

The authors added some comments on the text. However. they did not address one of the reviewer's biggest concerns enough. Unless the results are adjusted with potential other confounders such as injury pattern, I cannot believe the authors conclusion stating evidence from current practice in real world appears to support vertebroplasty as a viable choice for elderly people over 75 years old.  

Author Response

Dear Editors,

Thank you very much for your kind advice on my manuscript. My colleagues and I have made revisions according to these advices and highlighted in yellow.

If there is still revision needed, please feel free to let me know.

Merry Christmas and Happy New Year.

Jung-Der Wang, M.D., Sc.D.

Some minor changes would be required:

The conclusion in the abstract should say:

"Conclusion: Although conventional open surgery would usually be the best choice for the treatment of patients with pathological thoracolumbar fractures, database information from current practice in real world appears to support vertebroplasty as a viable choice for elderly people over 75 years old." (instead of "evidence", it should say "database information")

Response: Thanks. My colleagues and I agree that database information would be  more appropriate words and we have revised as follows: (Page 1, lines 28-31)

However, we were unable to rule out confounding by indication. Conclusion: Although conventional open surgery would usually be the best choice for the treatment of patients with pathological thoracolumbar fractures, database information from current practice in real world appears to support vertebroplasty as a viable choice for elderly people over 75 years old.

And section about conclusions in the manuscript should say: Conclusions
In Taiwan, patients aged over 75 with pathological thoracolumbar fractures receiving vertebroplasty had a better survival compared with those receiving conservative treatment. Moreover, vertebroplasty still showed improved survival in comparison with conservative treatment with an aHR of 0.87 after adjustment of age, sex, demographic and major clinical risk factors. Since we were unable to rule out confounding by indication, we conclude that the "overall" current practice of vertebroplasty in Taiwan seems acceptable or beneficial for long-term survival. More studies are needed to determine if vertebroplasty would improve quality of life.

(Include "overall" in the section Conclusions).

Response: Thanks. My colleagues and I agree that adding the term would be more appropriate and we have revised as follows: (Lines 233-235)

Since we were unable to rule out confounding by indication, we conclude that the overall current practice of vertebroplasty in Taiwan seems acceptable or beneficial for long-term survival. More studies are needed to determine if vertebroplasty would improve quality of life.